# DiaQ: Direction-aware Activation Quantization for Fast and Accurate Model Inference

## Abstract

How can we accelerate inference of matrix multiplications while maintaining the performance of neural networks? Weight-activation quantization reduces inference costs by quantizing both weights and activations, enabling cheaper matrix multiplications during inference. Previous researches on weight-activation quantization have focused on finding better weights to reduce quantization errors, while simply applying round-to-nearest (RTN) for the activations during inference. However, RTN has limitations in preserving the directional information of activations, which is crucial to accurately approximate matrix multiplications. In this paper, we propose DiaQ, an accurate method for quantizing activations while preserving directional information. DiaQ chooses the direction to round each value based on their direction as well as their distance from the quantization levels. DiaQ also extends each vector to prevent collapse during quantization and corrects the output scale to compensate for the change in magnitude after quantization. Extensive experiments show that DiaQ reduces the quantization error induced from activation quantization by up to 13.3% and 26.1% in terms of Euclidean and cosine distances, respectively, compared to RTN. DiaQ also improves the task performances of LLMs and ViTs.

## 1 Introduction

*How can we reduce the quantization error in matrix multiplication with activation quantization?* Recently, with the remarkable advancements in the field of artificial intelligence, the performance and size of deep models are continuously increasing (Kaplan et al., 2020; Chowdhery et al., 2023). As a result, demand for efficient inference is rising due to the growing inference costs. Weight-activation quantization is the most common approach to address this issue (Li et al., 2020; Hubara et al., 2021; Gholami et al., 2022). This technique represents weights and activations in low-bit integers during matrix multiplication, significantly reducing memory usage and inference time (Deng et al., 2020; Park et al., 2024).

Previous works on weight-activation quantization focus on finding better weights and activations to reduce the quantization error. For instance, SmoothQuant (Xiao et al., 2023) and OmniQuant (Shao et al., 2024) adjust the scales of weights and activations so that activations become easier to quantize. Moreover, recent studies such as QuaRot (Ashkboos et al., 2024), DuQuant (Lin et al., 2024a), and SpinQuant (Liu et al., 2025) further improve the quantizability of activations by applying rotation before quantization.

However, these works overlook how to effectively quantize activations and simply apply the RTN (round to nearest) for quantization (Ashkboos et al., 2024; Li et al., 2023). This leads to limitation in preserving the directional information of the activations. For instance, consider quantizing a vector $x = (7.3, 5.7)$ to integer values as shown in Figure 1. If we simply apply RTN, we would round 7.3 to 7 and 5.7 to 6, losing the original direction of the vector as shown in Figure 1 (b). On the other hand, if we round 7.3 to 8 and 5.7 to 6, the quantized vector preserves the direction of the original vector, as shown in Figure 1 (c).

To address this limitation, we propose Direction-aware Activation Quantization (DiaQ), an accurate activation quantization method that considers the direction of the activations during quantization to reduce the quantization error. DiaQ searches for the quantization level that preserves the direction of the original vector using direction-aware rounding. In this process, DiaQ extends the activation

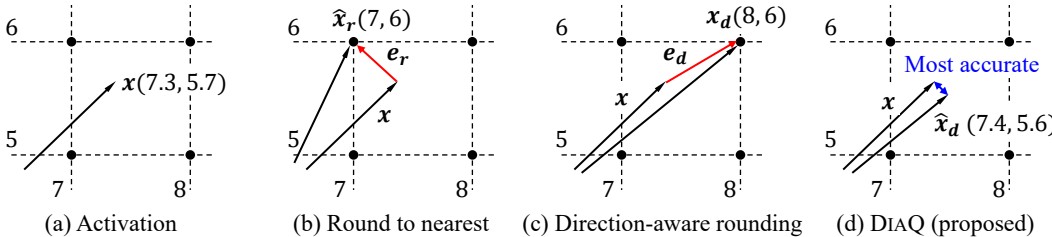

Figure 1: Illustration of the different schemes to round the vector $x$ (the proportions are exaggerated). When applying RTN as in (b), the error magnitude is smaller, but rounding to align the direction as in (c) reduces the cosine distance.

vector before quantization to prevent quantized vectors from collapsing to the origin. Finally, DIAQ scales the output of the quantized matrix multiplication to compensate for the change in magnitude during quantization.

Our main contributions are summarized as follows:

- **Theory.** We formally define the direction-aware approximation of a vector and prove that it reduces the approximation error of matrix-vector multiplication compared to RTN.
- **Algorithm.** We propose DIAQ, a new activation quantization algorithm to implement the direction-aware approximation in practice. DIAQ preserves the directional information of activations during quantization with the minimal computational overhead compared to matrix multiplication.
- **Experiment.** We show that DIAQ reduces the quantization error induced from activation quantization by up to 13.3% and 26.1% in terms of Euclidean and cosine distances, respectively, compared to RTN through extensive experiments. We also show that DIAQ improves the task performances of LLMs and ViTs. Our codes are available within the supplementary materials.

## 2 PRELIMINARIES

### 2.1 PROBLEM DEFINITION

Linear transforms are the core operations and the main computational bottleneck in various neural network architectures (Popescu et al., 2009; Gardner & Dorling, 1998; Vaswani et al., 2017; Tolstikhin et al., 2021). A linear transform is a set of matrix-vector multiplications, where an input vector $x$ is multiplied by a weight matrix $W$ to produce an output vector $y = Wx$. In this paper, we focus on efficient approximation of matrix-vector multiplications using quantization to accelerate neural network inference. We formally define the problem as follows.

**Problem 1** (Quantized Matrix Vector Multiplication). *Given a weight matrix $W \in \mathbb{R}^{m \times n}$, an input vector $x \in \mathbb{R}^n$, and a bit-width $b$, the goal is to approximate the output $\widehat{y} \approx Wx$ using $b$-bit integer operations with quantized weight $\widehat{W}$ and quantized input vector $\widehat{x}$, while minimizing the approximation error $\|\widehat{y} - Wx\|_2$.*

### 2.2 WEIGHT-ACTIVATION QUANTIZATION

Weight-activation quantization is a promising approach to efficiently approximate matrix-vector multiplications (Li et al., 2020; 2023; Ashkboos et al., 2024; Lin et al., 2024a; Liu et al., 2025). This method replaces full-precision matrix multiplications with lower-bit integer operations by quantizing both the weight matrix and the input vector to low-bit integer representations. The weight matrix $W$ is expressed as $\widehat{W} = s_w W_q$, where $W_q \in \mathbb{I}^{m \times n}$ is the quantized integer matrix, and $s_w$ is the scaling factor. The input vector $x$ is similarly quantized as $\widehat{x} = s_x x_q$, where $x_q \in \mathbb{I}^n$ is the quantized integer vector, and $s_x$ is the scaling factor. Then, the matrix-vector multiplication is approximated as $Wx \approx \widehat{W}\widehat{x} = s_w s_x (W_q x_q)$. This allows the matrix-vector multiplication to be performed using low-bit integer operations, which are significantly faster and more memory-efficient than full-precision operations (Tseng et al., 2024; Zhao et al., 2024; Lin et al., 2024c). Note that we ignore the zero-point and fuse it into the quantized integer matrices for simplicity in this paper, since it does not change the mathematical properties of the quantization error.

The weight matrix is quantized offline before inference, allowing users to search for the optimal quantization parameters (Xiao et al., 2023; Shao et al., 2024) and quantized weights (Frantar et al., 2023; Lin et al., 2024b; Nagel et al., 2020) without time constraints. On the other hand, the input vector is quantized online during inference since the input changes with each query. Thus, the input vector must be quantized quickly with minimal overhead to maintain fast inference speed.

The most common method for online activation quantization is the round-to-nearest (RTN) scheme (Gupta et al., 2015). This method first determines the quantization levels by finding the scaling factor $s_x$. This is obtained either 1) online by computing them per input token during inference, or 2) offline by pre-computing them using the activation statistics (e.g., min/max or percentile range) collected from a small calibration set. Then, each element of the input vector is rounded to the nearest quantization level, ensuring that the Euclidean distance between the original input vector $x$ and the approximated vector $\widehat{x}$ is minimized. Specifically, given an input vector $x$ and the scaling factor $s_x$, the input vector is approximated as follows:

$$\widehat{x} = s_x \left\lfloor \frac{x}{s_x} + \frac{1}{2} \right\rfloor. \tag{1}$$

## 3 Theoretical Analysis on Activation Quantization

How can we accurately approximate the product of a matrix $W$ and a vector $x$ by approximating the vector $x$ as $\widehat{x}$? A vector contains directional and magnitude information, both of which are crucial for accurate approximation. Previous works use the RTN (round to nearest) method, which approximates a given vector to the nearest quantization level. However, this method has limitations to preserve the directional information. This is because the direction of the error is formed independently of the original vector, as shown in Figure 1 (b).

To address this issue, we propose a direction-aware approximation method that considers the direction of the vector. First, we find the quantization level that has the highest cosine similarity with the given vector to preserve the directional information of $x$, as shown in Figure 1 (c). However, this distorts the magnitude information of the vector, as it prioritizes preserving the directional information over minimizing the absolute error. To compensate for this, we apply scaling to the quantized vector so that its magnitude matches that of the original vector, as shown in Figure 1 (d). In the remaining section, we theoretically prove that this approximation method is superior to the existing quantization method.

First, we establish Theorem 1 to set a criterion for better quantization methods. Theorem 1 states that reducing the approximation error of the vector leads to a reduction in the error of the matrix-vector multiplication itself. Hence, we need to find a quantization method that minimizes the approximation error of the vector.

**Theorem 1.** *For two approximations $\widehat{x}_1$ and $\widehat{x}_2$ of a given vector $x \in \mathbb{R}^n$, let $\|x - \widehat{x}_1\|_2 < \|x - \widehat{x}_2\|_2$. Then, for a matrix $W \in \mathbb{R}^{m \times n}$ following Gaussian distribution $\mathcal{N}(0, 1)^{m \times n}$, $\mathbb{E}(\|Wx - W\widehat{x}_1\|_2) < \mathbb{E}(\|Wx - W\widehat{x}_2\|_2)$.*

*Proof.* Let $e_1 = x - \widehat{x}_1$ and $e_2 = x - \widehat{x}_2$. Then, $Wx - W\widehat{x}_1 = We_1$ and $Wx - W\widehat{x}_2 = We_2$. Since each row of $W$ is independent and follows a Gaussian distribution, we have $We_1 \sim \mathcal{N}(0, \|e_1\|_2^2)^m$ and $We_2 \sim \mathcal{N}(0, \|e_2\|_2^2)^m$. Thus, $\|We_1\|_2 \sim \|e_1\|_2 \chi_m$ and $\|We_2\|_2 \sim \|e_2\|_2 \chi_m$, where $\chi_m$ is the Chi distribution with $m$ degrees of freedom. Therefore, $\mathbb{E}(\|Wx - W\widehat{x}_1\|_2) = \mathbb{E}(\|We_1\|_2) = \|e_1\|_2 \mathbb{E}(\chi_m) < \|e_2\|_2 \mathbb{E}(\chi_m) = \mathbb{E}(\|Wx - W\widehat{x}_2\|_2)$. □

Next, we formally define RTN and direction-aware approximation as Definitions 1 and 2, respectively.

**Definition 1** (RTN approximation). *For a given vector $x$, scale factor $s$, and quantization levels $\mathcal{Q} = \prod_{i=1}^{n}\{s\lfloor x_i/s \rfloor, s\lceil x_i/s \rceil\}$ near $x$, the RTN approximation $\widehat{x}_r$ of $x$ is $\widehat{x}_r = \arg\min_{l \in \mathcal{Q}} \|x - l\|_2$.*

**Definition 2** (Direction-aware approximation). *For a given vector $x$, scale factor $s$, and quantization levels $\mathcal{Q} = \prod_{i=1}^{n}\{s\lfloor x_i/s \rfloor, s\lceil x_i/s \rceil\}$ near $x$, the direction-aware approximation $\widehat{x}_d$ of $x$ is $\widehat{x}_d = \frac{\|x\|_2}{\|x_d\|_2} x_d$, where $x_d = \arg\max_{l \in \mathcal{Q}} \frac{x^\top l}{\|x\|_2 \|l\|_2}$.*

We now prove that the direction-aware approximation method reduces the approximation error of the vector more effectively than the RTN method. First, we present Theorem 2 to show that the

direction-aware approximation method has a smaller approximation error when the angle between $\boldsymbol{x}$ and $\widehat{\boldsymbol{x}}_d$ is sufficiently small.

**Theorem 2.** *Let $\widehat{\boldsymbol{x}}_r$ and $\widehat{\boldsymbol{x}}_d$ be the vectors approximated by the RTN and the direction-aware approximation, respectively, for a given vector $\boldsymbol{x}$ and scale factor $s$. Then, if the angle $\theta_d$ between $\boldsymbol{x}$ and $\widehat{\boldsymbol{x}}_d$ is sufficiently small, $\|\boldsymbol{x} - \widehat{\boldsymbol{x}}_d\|_2 < \|\boldsymbol{x} - \widehat{\boldsymbol{x}}_r\|_2$.*

*Proof.* Let $X, \widehat{X}_r$, and $\widehat{X}_d$ be points such that $\boldsymbol{x} = \overrightarrow{OX}, \widehat{\boldsymbol{x}}_r = \overrightarrow{O\widehat{X}_r}$, and $\widehat{\boldsymbol{x}}_d = \overrightarrow{O\widehat{X}_d}$, as shown in Figure 2. Also, let $\theta_r = \angle XO\widehat{X}_r$ and $\theta_d = \angle XO\widehat{X}_d$. Then, $\overline{OX} = \overline{O\widehat{X}_d}$ by the definition of $\widehat{\boldsymbol{x}}_d$. Let $H_r$ and $H_d$ be the feet of the perpendiculars from $X$ onto $\widehat{\boldsymbol{x}}_r$ and $\widehat{\boldsymbol{x}}_d$, respectively. Then, $\|\boldsymbol{x} - \widehat{\boldsymbol{x}}_d\|_2 = \overline{X\widehat{X}_d} = 2r \sin \frac{\theta_d}{2}$ for $r = \|\boldsymbol{x}\|_2$. Since $\theta_d \ll 1$, $2r \sin \frac{\theta_d}{2} \approx r \sin \theta_d$. Meanwhile, $\|\boldsymbol{x} - \widehat{\boldsymbol{x}}_r\|_2 = \overline{X\widehat{X}_r} \geq \overline{X\widehat{H}_r} = r \sin \theta_r$ since $\angle XH_r\widehat{X}_r = 90°$. Note that $\theta_d < \theta_r$ by the definition of $\widehat{\boldsymbol{x}}_d$: hence, we have $\|\boldsymbol{x} - \widehat{\boldsymbol{x}}_r\|_2 \geq r \sin \theta_r > r \sin \theta_d \approx \|\boldsymbol{x} - \widehat{\boldsymbol{x}}_d\|_2$. $\square$

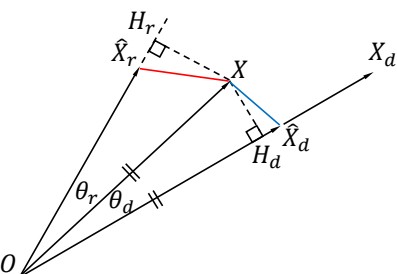

Figure 2: Illustration for Theorem 2.

We then prove Theorem 3, which shows that $\widehat{\boldsymbol{x}}_d$ reduces the approximation error of the vector more effectively than $\widehat{\boldsymbol{x}}_r$ using Theorem 2.

**Theorem 3.** *For a given vector $\boldsymbol{x}$ and scale factor $s$, let $\widehat{\boldsymbol{x}}_r$ and $\widehat{\boldsymbol{x}}_d$ be the vectors approximated by the RTN and the direction-aware approximation, respectively. Then, $\mathbb{E}(\|\boldsymbol{x} - \widehat{\boldsymbol{x}}_d\|_2) < \mathbb{E}(\|\boldsymbol{x} - \widehat{\boldsymbol{x}}_r\|_2)$.*

*Proof.* By Theorem 2, as the angle $\theta_d$ between $\boldsymbol{x}$ and $\widehat{\boldsymbol{x}}_d$ decreases, the approximation error of $\widehat{\boldsymbol{x}}_d$ becomes smaller than that of $\widehat{\boldsymbol{x}}_r$. Meanwhile, as the length of $\boldsymbol{x}$ increases, $\theta_d$ decreases (details in Appendix A.1). Thus, if we divide cases based on the quantization level where $\boldsymbol{x}$ lies, the worst case is when the quantization level touches the origin (details in Appendix A.2). Therefore, it suffices to prove only when $\mathcal{Q} = \prod_{i=1}^{n}\{\lfloor x_i/s \rfloor s, \lceil x_i/s \rceil s\}$ includes the origin.

Without loss of generality, consider the case where $\mathcal{Q} = \{0, s\}^n$. We assume that $\boldsymbol{x}$ is uniformly distributed in the space inside $\mathcal{Q}$ (Lin et al., 2024b). Hence, the quantization error of RTN follows a uniform distribution $\mathcal{U}(-0.5s, 0.5s)$. Thus, $\widehat{\boldsymbol{x}}_r - \boldsymbol{x} \sim \mathcal{U}(-0.5s, 0.5s)^n$, and the expected approximation error $\|\widehat{\boldsymbol{x}}_r - \boldsymbol{x}\|$ is $\mathbb{E}(\|\widehat{\boldsymbol{x}}_r - \boldsymbol{x}\|_2) = \frac{\sqrt{n}}{2\sqrt{3}}s \approx 0.289\sqrt{n}s$.

Now we calculate the expected value of the approximation error $\|\widehat{\boldsymbol{x}}_d - \boldsymbol{x}\|$ using the direction-aware approximation. Cosine similarity between $\boldsymbol{x}$ and $\boldsymbol{l} = (l_1, \cdots, l_n) \in \mathcal{Q}$ is $(\sum_{i \in \mathcal{S}} x_i)/(\sqrt{|\mathcal{S}|}\|\boldsymbol{x}\|_2)$, where $\mathcal{S} = \{i \mid l_i = s\}$. Thus, the maximum cosine similarity is $(\sum_{i=1}^{k} x^{(i)})/(\sqrt{k}\|\boldsymbol{x}\|_2)$ for $k = \sum_{i \in \mathcal{S}}$. Here, $x^{(1)}, \cdots, x^{(n)}$ are $x_1, \cdots, x_n$ sorted in descending order. Then, for the angle $\theta_d$ between $\boldsymbol{x}$ and $\widehat{\boldsymbol{x}}_d$, $\cos \theta_d = \max_{1 \leq k \leq n}(\sum_{i=1}^{k} x^{(i)})/(\sqrt{k}\|\boldsymbol{x}\|_2)$. Therefore, the expected value of $\cos \theta_d$ is $\frac{2\sqrt{2}}{3}$ when $k = \frac{2}{3}n$ (details in Appendix A.3). Thus, the expected error is $\mathbb{E}(\|\widehat{\boldsymbol{x}}_d - \boldsymbol{x}\|_2) = \mathbb{E}(2\|\boldsymbol{x}\|_2 \sin \frac{\theta_d}{2}) = (2\sqrt{n}s)/(\sqrt{3})\mathbb{E}(\sin \frac{\theta_d}{2}) \approx 0.195\sqrt{n}s$.

Therefore, even in the worst-case scenario for the direction-aware approximation, the expected approximation error is smaller than that of the RTN, i.e., $\mathbb{E}(\|\boldsymbol{x} - \widehat{\boldsymbol{x}}_d\|_2) < \mathbb{E}(\|\boldsymbol{x} - \widehat{\boldsymbol{x}}_r\|_2)$. $\square$

## 4 PROPOSED METHOD

In this section, we propose **DIAQ** (Direction-aware Activation Quantization), an accurate activation quantization algorithm to implement direction-aware approximation in Section 3.

### 4.1 OVERVIEW

We address the following challenges to implement direction-aware activation quantization:

**C1**. (**Collapsing to origin**) Most activation vectors are located near the origin, leading them to collapse toward zero during rounding. How can we prevent activations from collapsing?

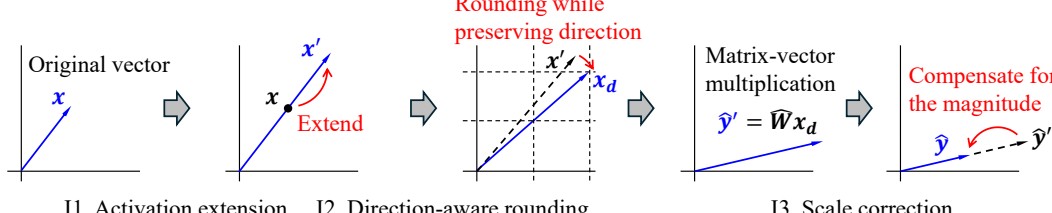

Figure 3: Overall process of DIAQ. Blue vectors denote the output of each step.

**C2**. (**Intractable search space**) Searching all quantization levels near the vector is computationally expensive. How can we efficiently find the quantization level with a similar direction?

**C3**. (**Distorted vector magnitudes**) Activations should be on quantization levels during matrix multiplication for efficient computation. How can we correct distortions in vector magnitudes while keeping them on quantization levels during multiplication?

To tackle these challenges, DIAQ exploits the following main ideas:

**I1**. (**Activation extension**) DIAQ extends the magnitude of each activation vector to prevent collapse while preserving its direction.

**I2**. (**Direction-aware rounding**) DIAQ rounds a vector considering not only its position but also its direction to find a quantization level aligned with the vector.

**I3**. (**Scale correction**) DIAQ assesses changes in vector magnitude during quantization, and corrects their scale after matrix multiplication.

Figure 3 shows the overall matrix multiplication process using DIAQ. DIAQ is composed of two pre-processing steps before matrix multiplication and one post-processing step after matrix multiplication. Before matrix multiplication, DIAQ first extends the activation vector to prevent collapse during quantization (Section 4.2). Then, DIAQ quantizes the extended activation using direction-aware rounding to preserve its directional information (Section 4.3). After matrix multiplication, DIAQ corrects the scale of the output to compensate for the change in magnitude during quantization (Section 4.4).

## 4.2 ACTIVATION EXTENSION

How can we prevent activations from collapsing toward the origin during quantization? Activations of neural networks such as LLMs and ViTs are concentrated around zero since they follow a Gaussian-like distribution or a power-law distribution (Yuan et al., 2022; Li et al., 2023; Ashkboos et al., 2024). Hence, directly quantizing the activation causes them to collapse toward the origin, resulting in the complete loss of directional information.

Our idea to prevent collapse is to move each vector away from the origin while preserving its direction. To achieve this, DIAQ extends the length of each activation by a fixed amount before quantization. For an activation vector $x$, quantization scale $s$, and an extension hyperparameter $\alpha$, DIAQ extends the length of $x$ by $\alpha$ relative to the quantization scale $s$ as shown in Figure 4(a) as follows:

$$x' = x + \alpha s \frac{x}{\|x\|_2} \tag{2}$$

Note that $s \frac{x}{\|x\|_2}$ is $x$ normalized to have the magnitude of $s$, a single step of quantization levels. Then, DIAQ quantizes the extended activation $x'$ instead of $x$ in the following step (Section 4.3).

## 4.3 DIRECTION-AWARE ROUNDING

How can we find a quantization level that aligns with the direction of a given activation vector $x' \in \mathbb{R}^n$ and quantization scale $s$? There are $2^n$ quantization levels surrounding $x'$ since we have two choices for each axis: rounding up or rounding down. Hence, naively computing the cosine similarity for all quantization levels around $x'$ is computationally infeasible as it requires evaluating $2^n$ candidates. Therefore, we need an efficient method to find a quantization level with high cosine similarity to $x'$ without explicitly calculating the cosine similarity.

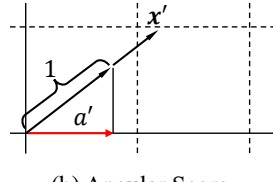 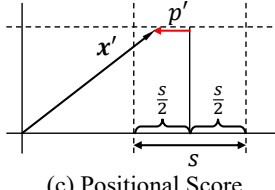

| (a) Activation Extension | (b) Angular Score | (c) Positional Score |

Figure 4: Illustrations of pre-processing steps of DIAQ. (a) DIAQ first extends the length of $x$ by $\alpha s$ to obtain $x'$. Then, to determine the rounding direction for the horizontal axis, (b) DIAQ computes the angular score with the horizontal component $a'$ of the normalized $x'$, and (c) the positional score with the signed distance $p'$ to $x'$ from the midpoint between quantization levels.

Then, how can we determine which of rounding up or rounding down for each axis yields the higher cosine similarity without directly computing the cosine similarity? Consider an error $e = l - x'$ between a given vector $x'$ and a quantization level $l$. The cosine similarity between $x'$ and $l$ increases as 1) the angle between $x'$ and $e$ decreases, and 2) the magnitude of $e$ decreases. Thus, we need to search for a quantization level $l$ that is 1) in the direction of $x'$ and 2) close to $x'$.

To jointly optimize these two criteria, we quantitatively evaluate angular score and positional score based on each criterion and combine them to determine the rounding direction for each axis. We score positive for rounding up and negative for rounding down, where the magnitude of the score indicates the strength of preference. We balance the expected magnitudes of the two scores for random input $x'$ to make them comparable. Let $a_i$ be the angular score and $p_i$ be the positional score for the $i$-th element $x'_i$ of a given vector $x'$. The total score $t_i$ to determine the rounding direction for the axis is defined as follows:

$$t_i = \beta a_i + p_i, \tag{3}$$

where $\beta$ is a balancing hyperparameter to adjust the importance between the angular score and the positional score. Then, we round up for the $i$-th axis if $t_i$ is positive, and round down if $t_i$ is negative.

Angular score $a_i$ represents the direction of $x'$ in the $i$-th axis. Hence, we use the $i$-th element $a'_i = x'_i/\|x'\|_2$ of the normalized vector $x'/\|x'\|_2$ to define the angular score as shown in Figure 4(b). This element is positive (negative) when $x'$ points in the positive (negative) direction of the $i$-th axis, so rounding up (down) is preferred. Note that rounding solely based on the angular score is equivalent to rounding up for positive elements and rounding down for negative elements, which maximizes the cosine similarity if positional effect is ignored (see Appendix A.4). Since the scale of each element of the normalized vector for random input $x' \in \mathbb{R}^n$ is on the order of $1/\sqrt{n}$ (see Appendix A.5), we define the angular score $a_i$ for the $i$-th axis by scaling $a'_i$ by $\sqrt{n}$ to ensure that the expected score becomes 1 as follows:

$$a_i = \sqrt{n}a'_i = \sqrt{n}\frac{x'_i}{\|x'\|_2}. \tag{4}$$

Positional score $p_i$ represents how close $x'_i$ is to either its rounded-up or rounded-down value. Hence, we use the signed distance $p'_i$ from the midpoint between rounded-up and rounded-down values to $x'_i$ to define the positional score as shown in Figure 4(c). When the distance is positive (negative), $x'_i$ positions right (left) of the midpoint, so rounding up (down) is preferred. Since the midpoint is $(\lfloor \frac{x'_i}{s} \rfloor + \frac{1}{2})s$, the signed distance $p'_i$ is $x'_i - s(\lfloor \frac{x'_i}{s} \rfloor + \frac{1}{2})$. This is distributed in the range of $[-\frac{s}{2}, \frac{s}{2}]$ so its expected magnitude for the random input $x'$ is $\frac{s}{4}$. Thus, we define the positional score $p_i$ for the $i$-th axis by scaling $p'_i$ by $\frac{4}{s}$, so that its scale matches with the angular score, as follows:

$$p_i = \frac{4}{s}p'_i = \frac{4}{s}\left(x'_i - s\left(\lfloor \frac{x'_i}{s} \rfloor + \frac{1}{2}\right)\right) \tag{5}$$

### 4.4 SCALE CORRECTION

How can we correctly obtain the direction-aware approximation $\widehat{y} = \widehat{W}\widehat{x}_d$ of the product of a quantized weight $\widehat{W}$ and an activation $x$ as in Definition 2? For a given activation vector $x$, DIAQ obtains a quantized activation $x_d$ by activation extension and direction-aware rounding. However, the

magnitude $\|\boldsymbol{x}_d\|_2$ of $\boldsymbol{x}_d$ is distorted from $\|\boldsymbol{x}\|_2$, while we need to multiply the quantized weight $\widehat{\boldsymbol{W}}$ with $\widehat{\boldsymbol{x}}_d = \frac{\|\boldsymbol{x}\|_2}{\|\boldsymbol{x}_d\|_2}\boldsymbol{x}_d$ whose magnitude is identical to $\boldsymbol{x}$ to accurately approximate the output. Directly scaling $\boldsymbol{x}_d$ would move it off the quantization levels, preventing efficient computation using integer operations.

Our idea to address this issue is to scale the output after quantized matrix-vector multiplication. In this way, we obtain the correctly approximated output while efficiently processing the multiplication using integer operations. DIAQ first obtains the output $\widehat{\boldsymbol{y}}' = \widehat{\boldsymbol{W}}\boldsymbol{x}_d$ with lower-bit integer operations. Then, DIAQ scales $\widehat{\boldsymbol{y}}'$ to obtain the correct approximation $\widehat{\boldsymbol{y}}$ as follows:

$$\widehat{\boldsymbol{y}} = \widehat{\boldsymbol{W}}\widehat{\boldsymbol{x}}_d = \widehat{\boldsymbol{W}}\left(\frac{\|\boldsymbol{x}\|_2}{\|\boldsymbol{x}_d\|_2}\boldsymbol{x}_d\right) = \frac{\|\boldsymbol{x}\|_2}{\|\boldsymbol{x}_d\|_2}\left(\widehat{\boldsymbol{W}}\boldsymbol{x}_d\right) = \frac{\|\boldsymbol{x}\|_2}{\|\boldsymbol{x}_d\|_2}\widehat{\boldsymbol{y}}'. \tag{6}$$

### 4.5 COMPLEXITY ANALYSIS

We analyze the computational complexity of DIAQ as Theorem 4.

**Theorem 4.** *The computational complexity of matrix-vector multiplication using* DIAQ *is* $O(mn)$, *given a weight matrix* $\boldsymbol{W} \in \mathbb{R}^{m \times n}$ *and an activation vector* $\boldsymbol{x} \in \mathbb{R}^n$.

*Proof.* The cost of the matrix multiplication is $O(mn)$, which remains unchanged. To analyze the additional cost incurred by DIAQ, we need to compute the costs of activation extension, direction-aware rounding, and scale correction.

To extend an activation, DIAQ first computes the length of the activation vector and then performs element-wise scaling. Computing the length of an activation vector requires summing the squares of each element, which takes $O(n)$ operations. The element-wise scaling operation also takes $O(n)$ operations. Direction-aware rounding involves computing the angular and positional scores for each element of the activation vector. Angular score is obtained by normalizing the activation vector, which requires $O(n)$ costs. Positional score is computed using element-wise rounding and subtraction, which also takes $O(n)$ costs. Finally, for scale correction, DIAQ compares the length of the quantized activation with that obtained during activation extension and performs element-wise scaling to correct the output. Similar to the activation extension process, this costs $O(n)$ as well. Since $O(n)$ costs are negligible compared to the cost of matrix multiplication, the total cost of matrix-vector multiplication remains $O(mn)$ even when using DIAQ. $\square$

## 5 EXPERIMENTS

We perform experiments to address the following questions.

**Q1**. **Error analysis (Section 5.2).** Does DIAQ reduce the quantization error compared to RTN?
**Q2**. **Hyperparameter analysis (Section 5.3).** How do the hyperparameters of DIAQ affect the quantization error?
**Q3**. **Task performance on LLMs (Section 5.4).** Does DIAQ improve the performance of the LLM?
**Q4**. **Task performance on ViTs (Section 5.5).** Does DIAQ improve the performance of the ViT?

### 5.1 EXPERIMENTAL SETUP

- **LLMs.** We use LLaMA-2 7B (Touvron et al., 2023) and LLaMA-3 8B (Dubey et al., 2024) models for LLMs. We quantize them with QuaRot (Ashkboos et al., 2024), and follow its implementation details. We use WikiText2 (Merity et al., 2017) dataset and ARC-Challenge, Arc-Easy (Clark et al., 2018), PIQA (Bisk et al., 2020), WinoGrande (Sakaguchi et al., 2021), and BoolQ (Clark et al., 2019) benchmarks to evaluate LLMs.
- **ViTs.** We use ViT (Wu et al., 2020), DeiT (Touvron et al., 2021) and Swin (Liu et al., 2021) model families for ViTs. We quantize them with RepQ-ViT (Li et al., 2023), and follow its implementation details. We use ImageNet (Deng et al., 2009) dataset to evaluate ViTs.
- **Linear layers.** We construct synthetic models with a single linear layer sampled from the Gaussian distribution. We apply the symmetric min-max quantization for the synthetic dataset. We use the synthetic inputs sampled from the Gaussian distribution to evaluate linear layers.

Table 1: Error analysis of activation quantization on various settings. $x$ denotes the activation, and $Wx$ represents the result of matrix multiplication. Euc. and Cos. refer to Euclidean distance and cosine distance, respectively. Refer to Appendix D for the complete results.

| Type | Model | Dataset | Method | $x$ Euc. ($\downarrow$) | $x$ Cos. ($\downarrow$) | $Wx$ Euc. ($\downarrow$) | $Wx$ Cos. ($\downarrow$) |
|---|---|---|---|---|---|---|---|
| LLM | Llama-3 8B | WikiText2 | RTN | 0.1508 | 0.0109 | 0.1238 | 0.0095 |
|  |  |  | DIAQ | **0.1341** | **0.0086** | **0.1098** | **0.0075** |
| ViT | ViT-B | ImageNet | RTN | 0.1718 | 0.0163 | 0.1016 | 0.0092 |
|  |  |  | DIAQ | **0.1511** | **0.0125** | **0.0881** | **0.0068** |
| Linear Layer | $\mathcal{N}^{4096\times4096}$ | $\mathcal{N}^{4096}$ | RTN | 0.1464 | 0.0106 | 0.1465 | 0.0106 |
|  |  |  | DIAQ | **0.1302** | **0.0085** | **0.1302** | **0.0085** |

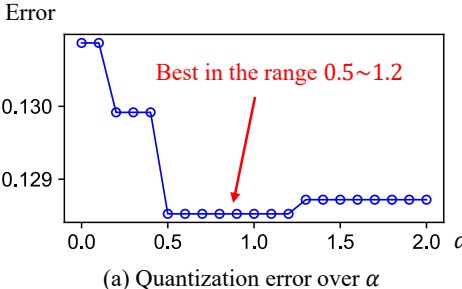

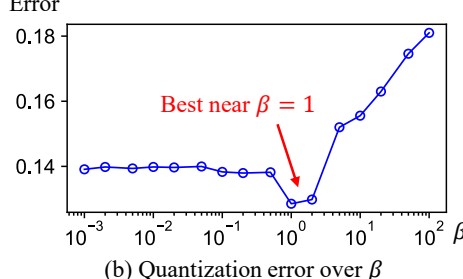

(a) Quantization error over $\alpha$      (b) Quantization error over $\beta$

Figure 5: Quantization error on the synthetic linear layers by varying the hyperparameters of DIAQ. (a) DIAQ achieves the lowest error when the extension hyperparameter $\alpha$ is in the range of 0.5 to 1.2. (b) DIAQ achieves the lowest error when the balancing hyperparameter $\beta$ is around 1.

## 5.2 ERROR ANALYSIS

We investigate the quantization errors during inference of quantized models. We observe the relative Euclidean distance and cosine distance from the original output for each matrix-vector multiplication in LLMs, ViTs, and linear layers when applying 4-bit quantization.

Table 1 shows the results of the experiments. DIAQ consistently reduces the quantization error of matrix-vector multiplication compared to RTN in terms of both Euclidean and cosine distances in all settings. DIAQ achieves up to 13.3% and 26.1% reduction in terms of Euclidean and cosine distances, respectively, on the output of matrix multiplication compared to RTN. This proves the effectiveness of preserving the directional information during activation quantization.

## 5.3 HYPERPARAMETER ANALYSIS

To analyze the effect of hyperparameters in DIAQ on quantization error, we observe the change of relative Euclidean error on the synthetic linear layers with 1024 dimensions by varying the hyperparameters. We also vary the extension hyperparameter $\alpha$ from 0 to 2 with an interval of 0.1 while fixing the balancing hyperparameter $\beta$ to 1. We vary the balancing hyperparameter $\beta$ from 0.001 to 100 in a logarithmic scale while fixing the extension hyperparameter $\alpha$ to 0.5.

Figure 5 shows the results of the experiments. As shown in Figure 5(a), DIAQ achieves the lowest quantization error when $\alpha$ is in the range of 0.5 to 1.2. This is because a small $\alpha$ does not sufficiently prevent the collapse of the vector, while a large $\alpha$ excessively extends the vector so that it exceeds the quantization range after clipping. Meanwhile, as shown in Figure 5(b), DIAQ achieves the lowest quantization error when $\beta$ is around 1. This is because a small $\beta$ makes DIAQ similar to RTN, which does not consider the directional information, while a large $\beta$ makes DIAQ to always round based on the sign, ignoring the distance from the quantization levels.

Table 2: Task accuracies of Llama models with DIAQ and RTN. PPL denotes perplexity, and AE, AC, PQ, WG, and BQ represent the zero-shot accuracy on ARC-Easy, ARC-Challenge, PIQA, WinoGrande, and BoolQ, respectively. Avg. refers to the average zero-shot accuracy.

| Model | Method | PPL ($\downarrow$) | Zero-shot accuracy ($\uparrow$) | | | | | |
| | | | AE | AC | PQ | WG | BQ | Avg. |
|---|---|---|---|---|---|---|---|---|
| Llama-2 7B | RTN | 6.13 | 68.6 | **42.3** | **77.2** | 64.5 | 73.0 | 65.1 |
| | DIAQ | **6.11** | **69.9** | 41.7 | 76.7 | **66.1** | **73.9** | **65.7** |
| Llama-3 8B | RTN | 8.17 | 69.5 | **45.6** | 75.3 | 67.1 | 74.6 | 66.4 |
| | DIAQ | **8.03** | **70.8** | 44.4 | **76.5** | **68.1** | **77.1** | **67.4** |

Table 3: Image classification accuracies of ViT models with DIAQ and RTN. We report the top-1 accuracy on ImageNet. Higher value indicates better performance.

| Bits | Method | ViT-S | ViT-B | DeiT-T | DeiT-S | DeiT-B | Swin-T | Swin-S |
|---|---|---|---|---|---|---|---|---|
| W6A6 | RTN | 80.43 | 83.62 | 70.76 | 78.90 | 81.27 | 80.69 | 82.79 |
| | DIAQ | **80.72** | **84.04** | **71.26** | **79.12** | **81.49** | **80.81** | **82.88** |
| W4A4 | RTN | 65.05 | 68.48 | 57.43 | 69.03 | 75.61 | 72.31 | 79.45 |
| | DIAQ | **65.65** | **69.78** | **59.32** | **69.53** | **76.24** | **72.70** | **79.81** |

## 5.4 TASK PERFORMANCE ON LLMS

To evaluate whether DIAQ improves the task accuracy of LLMs, we compress each model using QuaRot (Ashkboos et al., 2024), and measure perplexity and zero-shot reasoning accuracy. We use WikiText2 (Merity et al., 2017) to report the perplexity. We use ARC-Challenge, ARC-Easy (Clark et al., 2018), BoolQ (Clark et al., 2019), WinoGrande (Sakaguchi et al., 2021), and PIQA (Bisk et al., 2020) benchmarks to report the zero-shot reasoning accuracy using the language model evaluation harness (Gao et al., 2023). Table 2 shows the results of the experiments. DIAQ improves both perplexity and zero-shot reasoning accuracy compared to RTN in most cases, proving the effectiveness of preserving the directions during activation quantization for LLMs.

## 5.5 TASK PERFORMANCE ON VITS

To evaluate whether DIAQ improves the task accuracy of ViTs, we compress each model using RepQ-ViT (Li et al., 2023) and measure the top-1 accuracy on ImageNet (Deng et al., 2009). Table 3 summarizes the results of the experiments. DIAQ achieves higher accuracy than RTN in all cases, demonstrating the importance of directional information for ViTs.

## 6 CONCLUSION

We propose a direction-aware approximation scheme for matrix-vector multiplication and theoretically prove that it reduces the approximation error compared to the conventional RTN scheme. Our proposed DIAQ efficiently implements the direction-aware approximation. DIAQ performs direction-aware rounding to preserve the directional information during quantization. DIAQ also extends the activation vectors to prevent the quantized vectors from collapsing to the origin, and restores the magnitude change during quantization by scaling the output of the quantized matrix multiplication. Extensive experiments show that DIAQ effectively reduces the quantization error by up to 13.3% and 26.1% in terms of Euclidean and cosine distances, respectively, compared to RTN and improves task performance of LLMs and ViTs. These results indicate that unlike previous works that focus on modifying weights to absorb the difficulty of activation quantization, finding weights that enhance model performance would improve the performance combined with DIAQ. Future works include applying DIAQ during training quantized models to further improve the performance of weight-activation quantization.

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

# SUPPLEMENTARY MATERIALS

# A  DETAILS OF THEORETICAL ANALYSIS

## A.1  COSINE SIMILARITY FOR LONG ACTIVATIONS

**Lemma A.1.** *For given vectors $\boldsymbol{x}_1$, $\boldsymbol{x}_2$, and a scale factor $s$, let $\widehat{\boldsymbol{x}}_1$ and $\widehat{\boldsymbol{x}}_2$ be the vectors approximated by the direction-aware approximation for $\boldsymbol{x}_1$ and $\boldsymbol{x}_2$, respectively. Then, if $\|\boldsymbol{x}_1\| > \|\boldsymbol{x}_2\|$, $\mathbb{E}(sim(\boldsymbol{x}_1, \widehat{\boldsymbol{x}}_1)) \geq \mathbb{E}(sim(\boldsymbol{x}_2, \widehat{\boldsymbol{x}}_2))$ where $sim(\cdot)$ denotes the cosine similarity.*

Let $r_1 = \|\boldsymbol{x}_1\|$ and $r_2 = \|\boldsymbol{x}_2\|$ be the lengths of $\boldsymbol{x}_1$ and $\boldsymbol{x}_2$, respectively. Let $\mathcal{P}_1$ and $\mathcal{P}_2$ be the sets of quantization levels that can be rounded to from random vectors with lengths $r_1$ and $r_2$, respectively. Those are the quantization levels near the spherical shells with radii $r_1$ and $r_2$, respectively, as shown in Figure A.1. Then, $\widehat{\boldsymbol{x}}_1$ and $\widehat{\boldsymbol{x}}_2$ are the closest points to $\boldsymbol{x}_1$ and $\boldsymbol{x}_2$ in $\mathcal{P}_1$ and $\mathcal{P}_2$, respectively. Since $r_1 > r_2$, there are more points in $\mathcal{P}_1$ than in $\mathcal{P}_2$ because the spherical shell with radius $r_1$ is larger than that with radius $r_2$ so that the spherical shell with radius $r_1$ covers more area than that with radius $r_2$. Thus, the points of $\mathcal{P}_1$ are more densely distributed than those of $\mathcal{P}_2$ in angle as seen from the origin. Hence, we have $\mathbb{E}(\max_{\boldsymbol{l} \in \mathcal{P}_1} sim(\boldsymbol{x}_1, \boldsymbol{l})) \geq \mathbb{E}(\max_{\boldsymbol{l} \in \mathcal{P}_2} sim(\boldsymbol{x}_2, \boldsymbol{l}))$. Therefore, we obtain $\mathbb{E}(sim(\boldsymbol{x}_1, \widehat{\boldsymbol{x}}_1)) \geq \mathbb{E}(sim(\boldsymbol{x}_2, \widehat{\boldsymbol{x}}_2))$.

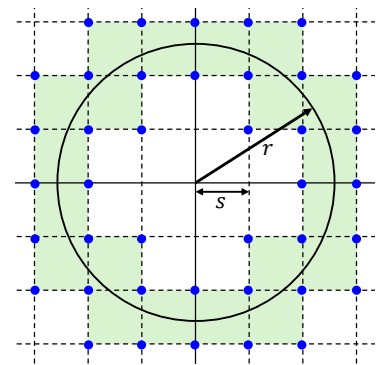

Figure A.1: Illustration of $\mathcal{P}$ for the given length $r$. Blue points denote the quantization levels in $\mathcal{P}$.

## A.2  CASES IN THE PROOF OF THEOREM 3

We present Figure A.2 to elaborate on the case partitioning in the proof of Theorem 3. As shown in Figure A.2(a), we divide cases based on where the vector $\boldsymbol{x}$ lies with respect to the quantization levels. Note that the angle $\theta_d$ between $\boldsymbol{x}$ and $\boldsymbol{x}_d$ becomes smaller as $\boldsymbol{x}$ moves away from the origin by Lemma A.1. Meanwhile, DIAQ outperforms RTN as $\theta_d$ becomes smaller by Theorem 2. Thus, the worst case of DIAQ compared to RTN occurs when $\boldsymbol{x}$ is located at one of the quantization levels closest to the origin, as shown in Figure A.2(b). Therefore, we prove Theorem 3 only for the case where the quantization level $\mathcal{Q} = \prod_{i=1}^{n}\{\lfloor x_i/s \rfloor s, \lceil x_i/s \rceil s\}$ includes the origin.

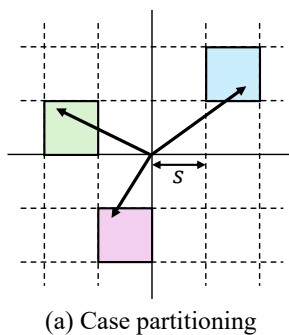

(a) Case partitioning

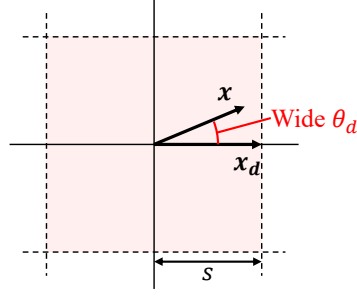

(b) Worst case of DiaQ

Figure A.2: Illustrations of the case partitioning in the proof of Theorem 3. Dotted lines represent the quantization levels.

## A.3  EXPECTED COSINE SIMILARITY

**Lemma A.2.** *For a given vector $\boldsymbol{x} \sim \mathcal{U}(0, s)^n$, where $\mathcal{U}(\cdot)$ denotes the uniform distribution, and quantization levels $\mathcal{Q} = \{0, s\}^n$ for a scale factor $s$, let $\boldsymbol{x}_d = \operatorname{argmax}_{\boldsymbol{l} \in \mathcal{Q}} \frac{\boldsymbol{x}^\top \boldsymbol{l}}{\|\boldsymbol{x}\|_2 \|\boldsymbol{l}\|_2}$ be the quantiza-*

*tion level with the highest cosine similarity with $\boldsymbol{x}$. Then, the expected cosine similarity between $\boldsymbol{x}$ and $\boldsymbol{x}_d$ is $\frac{2\sqrt{2}}{3}$.*

*Proof.* Cosine similarity between $\boldsymbol{x}$ and $\boldsymbol{l} = (l_1, \cdots, l_n) \in \mathcal{Q}$ is

$$\frac{\sum_{i \in \mathcal{S}} x_i}{\sqrt{|\mathcal{S}|}\|\boldsymbol{x}\|_2}, \tag{A.1}$$

where $\mathcal{S} = \{i \mid l_i = s\}$. Thus, the maximum cosine similarity when $|\mathcal{S}| = k$ is

$$\frac{\sum_{i=1}^{k} x^{(i)}}{\sqrt{k}\|\boldsymbol{x}\|_2}. \tag{A.2}$$

Here, $x^{(1)}, \cdots, x^{(n)}$ are $x_1, \cdots, x_n$ sorted in descending order. Then, for the angle $\theta_d$ between $\boldsymbol{x}$ and $\widehat{\boldsymbol{x}}_d$, we obtain

$$\cos\theta_d = \max_{1 \leq k \leq n} \frac{\sum_{i=1}^{k} x^{(i)}}{\sqrt{k}\|\boldsymbol{x}\|_2}. \tag{A.3}$$

Note that each $x_i$ is drawn from $\mathcal{U}(0, s)$. Hence, the expected sum of the top $k$ elements is

$$\mathbb{E}(\sum_{i=1}^{k} x^{(i)}) = \sum_{i=1}^{k}(1 - \frac{i}{n+1})s = (k - \frac{k(k+1)}{2(n+1)})s, \tag{A.4}$$

which is the sum of $k$ largest values among equally spaced $n$ values from $0$ to $s$.

Meanwhile, $\mathbb{E}(\|\boldsymbol{x}\|_2^2) = \sum_{i=1}^{n} \mathbb{E}(x_i^2) = \frac{n}{3}s^2$, so we have

$$\mathbb{E}(\|\boldsymbol{x}\|_2) = \sqrt{\frac{n}{3}}s. \tag{A.5}$$

Thus, we obtain the expected maximum cosine similarity as follows:

$$\mathbb{E}\left(\max_{1 \leq k \leq n} \frac{\sum_{i=1}^{k} x^{(i)}}{\sqrt{k}\|\boldsymbol{x}\|_2}\right) = \frac{(k - \frac{k(k+1)}{2(n+1)})}{\sqrt{kn/3}}. \tag{A.6}$$

Let $k = \rho n$ for $0 < \rho \leq 1$. Then, Equation (A.6) is approximated as follows:

$$\max_{\rho} \frac{n(\rho - \frac{\rho^2}{2})}{\sqrt{\rho n^2/3}} = \max_{\rho} \sqrt{3}\frac{\rho - \frac{\rho^2}{2}}{\sqrt{\rho}}. \tag{A.7}$$

This is maximized when $\rho = \frac{2}{3}$, and the maximum value is $\frac{2\sqrt{2}}{3}$. Therefore, the expected cosine similarity between $\boldsymbol{x}$ and $\boldsymbol{x}_d$ is $\frac{2\sqrt{2}}{3}$. $\qquad\square$

### A.4 SIGN-BASED ROUNDING

**Lemma A.3.** *For a given vector $\boldsymbol{x} = (x_1, \cdots, x_n) \in \mathbb{R}^n$ and a scale factor $s$, let $x_i^{(f)} = s\lfloor x_i/s \rfloor$ and $x_i^{(c)} = s\lceil x_i/s \rceil$ be the two quantization levels obtained by rounding down and up for $i$th element $x_i$, respectively. If we ignore the positional effect, i.e., $\|x_i - x_i^{(f)}\| = \|x_i - x_i^{(c)}\|$, then, the cosine similarity between $\boldsymbol{x}$ and a quantization level $\widehat{\boldsymbol{x}}$ is maximized when each positive element is rounded up and each negative element is rounded down.*

*Proof.* Let $\boldsymbol{e} = \widehat{\boldsymbol{x}} - \boldsymbol{x}$ be the quantization error vector. Then, $\boldsymbol{e} \in \{-\frac{s}{2}, \frac{s}{2}\}^n$ by the assumption. Thus, the length of $\boldsymbol{e}$ is fixed to $\|\boldsymbol{e}\|_2 = \frac{s}{2}\sqrt{n}$ so that the angle $\theta = \arccos(\frac{|\boldsymbol{x}^\top \boldsymbol{e}|}{\|\boldsymbol{x}\|_2\|\boldsymbol{e}\|_2})$ between $\boldsymbol{x}$ and $\boldsymbol{e}$ is narrowest when the numerator $\|\boldsymbol{x}^\top \boldsymbol{e}\|$ is maximized. Since all elements in $\boldsymbol{e}$ have the same magnitude $\frac{s}{2}$, $\|\boldsymbol{x}^\top \boldsymbol{e}\|$ is maximized when all elements in $\boldsymbol{e}$ have the same sign as those in $\boldsymbol{x}$ or have the opposite sign as those in $\boldsymbol{x}$. Hence, the quantization level $\widehat{\boldsymbol{x}}$ with the highest cosine similarity with $\boldsymbol{x}$ is either

$\widehat{\boldsymbol{x}}^{(+)} = (x_1 + \text{sign}(x_1)\frac{s}{2}, \cdots, x_n + \text{sign}(x_n)\frac{s}{2})$ or $\widehat{\boldsymbol{x}}^{(-)} = (x_1 - \text{sign}(x_1)\frac{s}{2}, \cdots, x_n - \text{sign}(x_n)\frac{s}{2})$. Note that $\|\widehat{\boldsymbol{x}}^{(+)}\|_2 > \|\widehat{\boldsymbol{x}}^{(-)}\|_2$ since each element in $\widehat{\boldsymbol{x}}^{(+)}$ has larger magnitude than that in $\widehat{\boldsymbol{x}}^{(-)}$. Thus, $\widehat{\boldsymbol{x}}^{(+)}$ is closer to $\boldsymbol{x}$ than the angle bisector between $\widehat{\boldsymbol{x}}^{(+)}$ and $\widehat{\boldsymbol{x}}^{(-)}$, as shown in Figure A.3. Hence, $\text{sim}(\boldsymbol{x}, \widehat{\boldsymbol{x}}^{(+)}) > \text{sim}(\boldsymbol{x}, \widehat{\boldsymbol{x}}^{(-)})$, where $\text{sim}(\cdot)$ denotes the cosine similarity. Therefore, the cosine similarity is maximized when each positive element is rounded up and each negative element is rounded down. $\qquad\square$

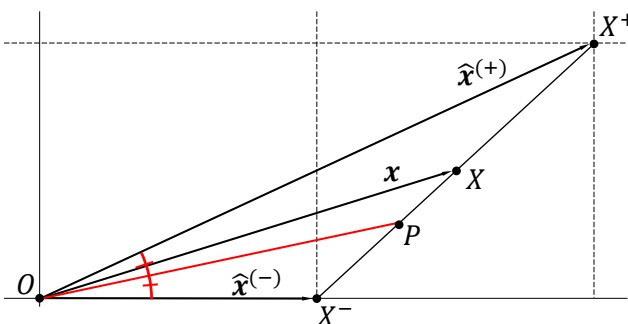

Figure A.3: Let $X$, $X^+$, and $X^-$ be points such that $\boldsymbol{x} = \overrightarrow{OX}, \widehat{\boldsymbol{x}}^{(+)} = \overrightarrow{OX^+}$, and $\widehat{\boldsymbol{x}}^{(-)} = \overrightarrow{OX^-}$, respectively. Then, $X$ is the midpoint of the $\overline{X^+X^-}$. Let $P$ be the foot of the angle bisector between $\overrightarrow{OX^+}$ and $\overrightarrow{OX^-}$ onto $\overline{X^+X^-}$. Then, $\overline{PX^+} > \overline{PX^-}$ since $\|\widehat{\boldsymbol{x}}^{(+)}\|_2 > \|\widehat{\boldsymbol{x}}^{(-)}\|_2$ and $\overline{PX^+} : \overline{PX^-} = \|\widehat{\boldsymbol{x}}^{(+)}\|_2 : \|\widehat{\boldsymbol{x}}^{(-)}\|_2$. Thus, $X$ is closer to $X^+$ than $P$ so that $\angle XOX^+ < \angle POX^+$.

### A.5 Expected Magnitude of Elements in Normalized Vector

For a random vector $\boldsymbol{x} \in \mathbb{R}^n$ drawn from a Gaussian distribution, let the normalized vector $\boldsymbol{u} = \frac{\boldsymbol{x}}{\|\boldsymbol{x}\|_2} = (u_1, \cdots, u_n)$. Then, we have $\sum_{i=1}^n u_i^2 = 1$ since $\|\boldsymbol{u}\|_2 = 1$. Thus, we have $\mathbb{E}(\sum_{i=1}^n u_i^2) = 1$. Since each element $u_i$ is identically distributed, we have $n\mathbb{E}(u_i^2) = 1$ for any $i$. Hence, we obtain $\mathbb{E}(u_i^2) = \frac{1}{n}$ so the scale of $u_i$ is on the order of $\frac{1}{\sqrt{n}}$.

## B Algorithm

Algorithm 1 summarizes the process of quantized matrix-vector multiplication with DIAQ. Note that the quantized matrix multiplication is equivalent to performing quantized matrix-vector multiplication for each column of the activation. From lines 1 to 2, DIAQ first obtains the quantization parameters of the activation $\boldsymbol{x}$ and preprocesses the activation $\boldsymbol{x}$ to be quantized. From lines 3 to 4, DIAQ extends the length of the activation $\boldsymbol{x}$ by $\alpha s$ and obtains $\boldsymbol{x}'$ to prevent the collapse during rounding. From lines 5 to 15, DIAQ performs direction-aware rounding on the extended activation $\boldsymbol{x}'$ and obtains the quantized activation $\boldsymbol{x}_d$. Note that the for loop is executed in parallel for each element. From lines 16 to 17, DIAQ multiplies the quantized activation $\boldsymbol{x}_d$ with the quantized weight $\widehat{\boldsymbol{W}}$ and corrects the magnitude of the result. Finally, DIAQ returns the output in line 18.

## C Details on the Experimental Setup

We present the detailed experimental settings in this section.

### C.1 Implementation Details

We modify the activation quantizers of the official implementation of QuaRot (https://github.com/spcl/QuaRot) and RepQ (https://github.com/zkkli/RepQ-ViT) for experiments. Specifically, we insert a pre-processing module to extend the activation and control the rounding direction before each quantizer, and add a post-processing step to scale the output after the matrix multiplication.

---

**Algorithm 1** Quantized matrix-vector multiplication using DIAQ

---

**Input:** quantized weight $\widehat{\boldsymbol{W}} \in \mathbb{R}^{m \times n}$, activation vector $\boldsymbol{x} \in \mathbb{R}^n$, extension hyperparameter $\alpha$, and balancing hyperparameter $\beta$.
**Output:** Approximated product $\widehat{\boldsymbol{y}}$ of the weight and activation.

/** **Step 0: setting up quantization parameters** **/
1: Compute the quantization scale $s$ for the activation $\boldsymbol{x}$ based on the quantization scheme.
2: Clip the activation $\boldsymbol{x}$ according to the quantization scheme.

/** **Step 1: activation extension** **/
3: $\boldsymbol{u} \leftarrow \frac{\boldsymbol{x}}{\|\boldsymbol{x}\|_2}$
4: $\boldsymbol{x}' \leftarrow \boldsymbol{x} + \alpha s \boldsymbol{u}$

/** **Step 2: direction-aware rounding** **/
5: $\boldsymbol{x}'^{(f)} \leftarrow s \lfloor \boldsymbol{x}'/s \rfloor, \ \boldsymbol{x}'^{(c)} \leftarrow s \lceil \boldsymbol{x}'/s \rceil$
6: $\boldsymbol{a} \leftarrow \sqrt{n}\boldsymbol{u}, \ \boldsymbol{p} \leftarrow \frac{4}{s} \left( \boldsymbol{x}' - (\boldsymbol{x}'^{(f)} + \frac{s}{2}\mathbf{1}) \right)$
7: $\boldsymbol{t} \leftarrow \beta \boldsymbol{a} + \boldsymbol{p}$
8: $\boldsymbol{x}_d \leftarrow \mathbf{0}$
9: **for** $i = 0$ to $n - 1$ **do**
10:     **if** $\boldsymbol{t}[i] > 0$ **then**
11:         $\boldsymbol{x}_d[i] \leftarrow \boldsymbol{x}'^{(c)}[i]$
12:     **else**
13:         $\boldsymbol{x}_d[i] \leftarrow \boldsymbol{x}'^{(f)}[i]$
14:     **end if**
15: **end for**

/** **Step 3: scale correction** **/
16: $\widehat{\boldsymbol{y}}' \leftarrow \widehat{\boldsymbol{W}} \boldsymbol{x}_d$
17: $\widehat{\boldsymbol{y}} \leftarrow \frac{\|\boldsymbol{x}\|_2}{\|\boldsymbol{x}_d\|_2} \widehat{\boldsymbol{y}}'$

18: **return** $\widehat{\boldsymbol{y}}$

---

## C.2 HYPERPARAMETER SEARCH

We report quantization errors when $\alpha = 0.5$ and $\beta = 1.0$. For other results, we search for the extension hyperparameter $\alpha$ in the range of $(0, 2]$. We search for the balancing hyperparameter $\beta$ in the range of $(0, 100]$. Each hyperparameter is searched up to one significant digit. We prioritize zero-shot accuracy over quantization error or perplexity when selecting hyperparameters for LLMs, as zero-shot benchmarks evaluate the model's capability of commonsense reasoning, which is more critical in practical use cases.

## C.3 QUANTIZATION ERROR

We measure the error induced by approximating the given vector $\boldsymbol{x}$ to $\widehat{\boldsymbol{x}}$ for each quantized matrix-vector multiplication with a weight $\boldsymbol{W}$ and an activation $\boldsymbol{x}$ as follows.

$$e_1 = \frac{\|\boldsymbol{x} - \widehat{\boldsymbol{x}}\|_2}{\|\boldsymbol{x}\|_2} \tag{C.8}$$

$$c_1 = 1 - \frac{\boldsymbol{x}^\top \widehat{\boldsymbol{x}}}{\|\boldsymbol{x}\|_2 \|\widehat{\boldsymbol{x}}\|_2} \tag{C.9}$$

$$e_2 = \frac{\|\boldsymbol{W}\boldsymbol{x} - \boldsymbol{W}\widehat{\boldsymbol{x}}\|_2}{\|\boldsymbol{W}\boldsymbol{x}\|_2} \tag{C.10}$$

$$c_2 = 1 - \frac{(\boldsymbol{W}\boldsymbol{x})^\top (\boldsymbol{W}\widehat{\boldsymbol{x}})}{\|\boldsymbol{W}\boldsymbol{x}\|_2 \|\boldsymbol{W}\widehat{\boldsymbol{x}}\|_2} \tag{C.11}$$

$\widehat{\boldsymbol{x}}$ is either $\widehat{\boldsymbol{x}}_r$ or $\widehat{\boldsymbol{x}}_d$ depending on whether $\boldsymbol{x}$ is approximated by RTN (Definition 1) or the proposed direction-aware approximation (Definition 2). $e_1$ and $c_1$ represent the normalized Euclidean error and the cosine error of the input vector, respectively. $e_2$ and $c_2$ denote the normalized Euclidean error and the cosine error of the output, respectively. We observe the quantization errors during measuring perplexity on WikiText2 for LLMs, and validating accuracy on ImageNet for ViTs. We report the average of all quantization errors obtained from the entire model.

### C.4 PERPLEXITY

We use WikiText2 (Merity et al., 2017) dataset to evaluate the perplexity of each model. We set the sequence length during the evaluation to 2048, which is the default settings of QuaRot.

### C.5 ZERO-SHOT EVALUATION

We use zero-shot evaluation benchmarks to assess the model's capability for commonsense reasoning. We use the following datasets for the evaluation.

- **ARC-Challenge and ARC-Easy** (Clark et al., 2018) are composed of grade-school level science problems. They are divided into challenge and easy categories based on whether they can be solved using naïve algorithms.
- **PIQA** (Bisk et al., 2020) consists of questions to choose a possible solution for the given physical scenario.
- **WinoGrande** (Sakaguchi et al., 2021) is a task to find a proper entity that a pronoun is representing in the given sentence.
- **BoolQ** (Clark et al., 2019) requires the model to answer in either yes or no to a question based on the given passage.

Table C.1 shows the statistics of benchmarks.

Table C.1: Statistics of zero-shot commonsense reasoning benchmarks.

| Dataset | Instance | Choices |
|---|---|---|
| ARC-Challenge | 1,172 | Multiple (4) |
| ARC-Easy | 2,376 | Multiple (4) |
| PIQA | 3,000 | Binary (2) |
| WinoGrande | 1,267 | Binary (2) |
| BoolQ | 3,270 | Binary (2) |

### C.6 IMAGE CLASSIFICATION

We use ImageNet (ILSVRC 2012) (Deng et al., 2009) dataset to evaluate the classification accuracy of ViTs. We follow the same evaluation protocol as RepQ (Li et al., 2023).

## D ADDITIONAL EXPERIMENTAL RESULTS

In this section, we present additional results of error analysis. Table D.2 shows the activation quantization errors for all models. Note that DIAQ consistently reduces the quantization error compared to RTN in all settings.

## E THE USE OF LARGE LANGUAGE MODELS

LLMs are used to facilitate code implementation, layout formatting, and the polishing of the writing in this paper.

Table D.2: Error analysis of activation quantization on various settings. $x$ denotes the activation, and $Wx$ represents the result of matrix multiplication. Euc. and Cos. refer to Euclidean distance and cosine distance, respectively.

| Type | Model | Dataset | Method | $x$ | | $Wx$ | |
|------|-------|---------|--------|-----------|-----------|-----------|-----------|
| | | | | Euc. ($\downarrow$) | Cos. ($\downarrow$) | Euc. ($\downarrow$) | Cos. ($\downarrow$) |
| LLM | Llama-2 7B | WikiText2 | RTN | 0.1392 | 0.0096 | 0.1068 | 0.0064 |
| | | | DIAQ | **0.1246** | **0.0078** | **0.0958** | **0.0052** |
| | Llama-3 8B | WikiText2 | RTN | 0.1508 | 0.0109 | 0.1238 | 0.0095 |
| | | | DIAQ | **0.1341** | **0.0086** | **0.1098** | **0.0075** |
| ViT | ViT-S | ImageNet | RTN | 0.1583 | 0.0133 | 0.1048 | 0.0083 |
| | | | DIAQ | **0.1419** | **0.0109** | **0.0953** | **0.0070** |
| | ViT-B | ImageNet | RTN | 0.1718 | 0.0163 | 0.1016 | 0.0092 |
| | | | DIAQ | **0.1511** | **0.0125** | **0.0881** | **0.0068** |
| | DeiT-T | ImageNet | RTN | 0.1444 | 0.0111 | 0.1149 | 0.0087 |
| | | | DIAQ | **0.1314** | **0.0094** | **0.1051** | **0.0074** |
| | DeiT-S | ImageNet | RTN | 0.1474 | 0.0116 | 0.0982 | 0.0074 |
| | | | DIAQ | **0.1333** | **0.0097** | **0.0899** | **0.0062** |
| | DeiT-B | ImageNet | RTN | 0.1533 | 0.0133 | 0.1061 | 0.0119 |
| | | | DIAQ | **0.1353** | **0.0102** | **0.0952** | **0.0101** |
| | Swin-T | ImageNet | RTN | 0.1507 | 0.0134 | 0.0959 | 0.0066 |
| | | | DIAQ | **0.1317** | **0.0101** | **0.0863** | **0.0057** |
| | Swin-S | ImageNet | RTN | 0.1550 | 0.0139 | 0.1223 | 0.0127 |
| | | | DIAQ | **0.1341** | **0.0103** | **0.1077** | **0.0099** |
| Linear Layer | $\mathcal{N}^{1024\times1024}$ | $\mathcal{N}^{1024}$ | RTN | 0.1318 | 0.0086 | 0.1319 | 0.0086 |
| | | | DIAQ | **0.1191** | **0.0071** | **0.1192** | **0.0072** |
| | $\mathcal{N}^{2048\times2048}$ | $\mathcal{N}^{2048}$ | RTN | 0.1396 | 0.0097 | 0.1398 | 0.0097 |
| | | | DIAQ | **0.1250** | **0.0079** | **0.1250** | **0.0078** |
| | $\mathcal{N}^{4096\times4096}$ | $\mathcal{N}^{4096}$ | RTN | 0.1464 | 0.0106 | 0.1465 | 0.0106 |
| | | | DIAQ | **0.1302** | **0.0085** | **0.1302** | **0.0085** |

