# OpenReview forum: "DiaQ: Direction-aware Activation Quantization for Fast and Accurate Model Inference"
_ICLR.cc/2026/Conference — ICLR 2026 Conference Withdrawn Submission_

### Official Review · Reviewer_1rae · 2025-10-17

**Soundness:** 1
**Presentation:** 1
**Contribution:** 1
**Rating:** 0
**Confidence:** 3

**Summary:**

This paper studies the problem of weight-activation quantization and presents a DIAQ method for quantizing activations while preserving
directional information.

**Strengths:**

I am not sure. The paper is difficult to decipher and has marginal quality in all aspects.

**Weaknesses:**

- Poor literary presentation.
- Lack of experimental comparison with SOTA methods.
- It is unclear to me what the authors meant by "direction-aware quantization" because direction is never precisely defined or provided a context for its usage.
- I have found most theorems and figures in this paper ad-hoc and lacking sufficient technical depth/novelty.
- I cannot understand how RTN can serve as a baseline method.

**Questions:**

I am sorry, but this paper does not make much sense to me. So it is difficult for me to ask meaningful questions.

---

### Official Review · Reviewer_6hZX · 2025-10-31

**Soundness:** 2
**Presentation:** 3
**Contribution:** 3
**Rating:** 2
**Confidence:** 4

**Summary:**

This article argues that traditional activation RTN quantization fails to ensure directional consistency, and thus proposes DiaQ. Specifically, it first increases the modulus of weights while keeping them in their original direction, then puts forward a direction-preserving quantization strategy, and finally eliminates the impact of modulus growth on the results of matrix multiplication. Experimentally, the authors applied DiaQ on the basis of QuaRot's activation quantization, which improved the quantization accuracy.

**Strengths:**

1. The article presents its research motivation very clearly, and Figure 1 is particularly well-designed.
2. The article raises a crucial research question: whether round-to-nearest quantization is truly the most suitable rounding method.
3. The article describes its methodology with exceptional clarity, and the formula derivations are complete.

**Weaknesses:**

My biggest question is: Weights are generally subject to static quantization, while activations rely on dynamic quantization, which thus imposes higher requirements on quantization efficiency. However, the article fails to provide a clear analysis or experimental comparison between RTN and DiaQ. If this issue can be clearly addressed, I believe this article is worthy of recommendation and holds significant importance for the field of quantization.

Other weaknesses:
1. The article emphasizes the need to preserve the direction of the original data but does not clearly explain the significance of doing so. Mere experimental comparison is not sufficiently persuasive; additional theoretical analysis is required.
2. The technical route involved in the article is scalar quantization, yet the authors treat the original data as vectors when emphasizing the concept of "direction," leading to conceptual confusion.
3. The experiments on LLM quantization are insufficient, and the model size is relatively limited (at the 7B level). Referring to QuaRot and other PTQ works based on rotation, quantization experiments on larger-scale models are essential.
4. Ablation experiments are lacking.

**Questions:**

1. ViT uses LayerNorm instead of RMSNorm; how is it adapted to QuaRot?
2. Is this method more suitable for weights in static quantization? And is it effective?

---

### Official Review · Reviewer_5Axc · 2025-10-31

**Soundness:** 3
**Presentation:** 3
**Contribution:** 2
**Rating:** 2
**Confidence:** 4

**Summary:**

The paper proposes DiaQ, a general post training activation quantization method for fast model inference. Rather than rounding to the nearest quantized value, the paper proposes to round such that the activation vector preserves its directional information. At its core, the method relies on two scoring functions, an angular score and a positional score. These scores are linearly combined to decide whether components of the activation vectors should be rounded up or down.

**Strengths:**

The paper is written in an understandable way. The core ideas are easy to follow.

**Weaknesses:**

- From the presentation, the method claims to be general, however experiments are only conducted for transformer models (ViT and Llama)
- Experiments only compare to the naive round to nearest (RTN) approach. This is simply not enough, since there are many other more advanced activation quantization methods. For example: https://arxiv.org/abs/2312.05693 gives another SOTA method for activation quantization for LLMs and compares to many other related baselines.

**Questions:**

The main motivation for your method is that the direction information of the activation vectors should be preserved as much as possible. However, you also argue that finding the quantized activation activation that preserves the direction best is computationally complex and impractical. Therefore, you introduce a heuristic consisting of activation extension, angular score and positional score in section 4.2 and 4.3.

Purely from the formulas, it is very hard to understand the dynamics of your quantization method. Can you give conditions when your method can recover the best direction preserving quantization? Or can you provide small scale experiments where you tried if (or how often) your heuristic could recover the best solution?

---

### Official Review · Reviewer_iCsX · 2025-11-01

**Soundness:** 1
**Presentation:** 2
**Contribution:** 2
**Rating:** 2
**Confidence:** 5

**Summary:**

The paper proposes a new quantization method that preserves the direction of activation vectors to reduce approximation error. Traditional RTN schemes minimize Euclidean distance but ignore directional information, which can distort model behavior. DIAQ introduces direction-aware rounding, selecting quantization levels that maximize cosine similarity with the original vector, followed by a scaling step to restore magnitude. Experiments show DIAQ decreases activation quantization error by up to 13.3% (Euclidean) and 26.1% (cosine) compared to RTN, and enhances performance across LLMs and ViTs.

**Strengths:**

(1) The core idea is interesting and intuitively appealing. By preserving the direction of high-dimensional activation vectors during quantization, DIAQ helps maintain the geometric structure of the data. This aspect could be particularly valuable in large-scale models where cosine similarity among feature vectors carries semantic meaning.
(2) The method appears computationally lightweight. Since it modifies the rounding and scaling process with minimal additional cost, it seems well-suited for on-demand activation quantization in LLM inference, where latency and efficiency are critical.

**Weaknesses:**

(1) Although the paper emphasizes direction preservation, it lacks a rigorous theoretical explanation for why this property necessarily leads to lower quantization error beyond intuitive geometric reasoning. The analysis provided does not fully quantify the causal relationship between direction maintenance and reduced inference error.
(2) Experimental comparisons are limited. The improvements over the RTN baseline are marginal, and stronger baselines, such as rotation-based or adaptive activation quantization methods, are missing. This raises doubts about whether DIAQ offers a meaningful methodological advantage or merely a small refinement.
(3) The paper claims practical benefits for fast inference but does not present system-level benchmarks or hardware latency evaluations. Without empirical runtime or throughput analysis, the claims of efficiency and scalability remain speculative, weakening the paper’s engineering credibility.

**Questions:**

Could you provide a stronger theoretical link between preserving vector direction and reducing quantization-induced output error? The current analysis focuses on vector approximation but does not clearly explain why directional alignment translates into better downstream model performance. A mathematical bound or intuition involving cosine similarity and activation propagation would be valuable.


How exactly is the direction-aware rounding implemented? Do you search over neighboring quantization levels for each element, or is it a closed-form adjustment? Clarifying the computational complexity relative to RTN would help assess practical deployability, especially for token-wise quantization in LLM inference. Can you share runtime or throughput benchmarks on real hardware (GPU/TPU) comparing DIAQ and RTN? Even small per-vector overhead can accumulate in large-scale inference. Such empirical evidence would make the “fast inference” claim more convincing.

Why does the paper only compare against RTN? Methods such as QuaRot, SmoothQuant, OmniQuant, or SpinQuant may already incorporate geometric preservation implicitly. Including these would situate DIAQ more clearly within the state-of-the-art landscape.

Does DIAQ interact with existing activation calibration techniques or post-training quantization pipelines? For example, does scaling interfere with percentile-based clipping or outlier smoothing? Could you visualize angular error distributions before and after applying DIAQ? This might help readers intuitively see how direction preservation affects quantization geometry.

---

### Note · Authors · 2025-11-13

**Comment:**

We sincerely appreciate the reviewers’ constructive feedback. We will use these insights to further improve and refine our work.

**Withdrawal Confirmation:**

I have read and agree with the venue's withdrawal policy on behalf of myself and my co-authors.